# Uncertainty in Rainfall Intensity Duration Frequency Curves of Peninsular Malaysia under Changing Climate Scenarios

**Muhammad Noor [1], Tarmizi Ismail [1], Eun-Sung Chung [2],*** **, Shamsuddin Shahid [1]** **and Jang Hyun Sung [3]**

[1]  Department of Water and Environmental Engineering, School of Civil Engineering, Universiti Teknologi Malaysia (UTM), Johor Bahru 81310, Malaysia; mnkakar@gmail.com (M.N.); tarmiziismail@utm.my (T.I.); sshahid@utm.my (S.S.)
[2]  Faculty of Civil Engineering, Department of Civil Engineering, Seoul National University of Science and Technology, Seoul 01811, Korea
[3]  Ministry of Environment, Han River Flood Control Office, Seoul 06501, Korea; jhsung1@korea.kr
*  Correspondence: eschung@seoultech.ac.kr; Tel.: +82-10-970-9017

**Abstract:** This study developed a methodological framework to update the rainfall intensity-duration-frequency (IDF) curves under climate change scenarios. A model output statistics (MOS) method is used to downscale the daily rainfall of general circulation models (GCMs), and an artificial neural network (ANN) is employed for the disaggregation of projected daily rainfall to hourly maximum rainfall, which is then used for the development of IDF curves. Finally, the 1st quartiles, medians, and 3rd quartiles of projected rainfall intensities are estimated for developing IDF curves with uncertainty level. Eight GCM simulations under two radiative concentration pathways (RCP) scenarios, namely, RCP 4.5 and RCP 8.5, are used in the proposed framework for the projection of IDF curves with related uncertainties for peninsular Malaysia. The projection of rainfall revealed an increase in the annual average rainfall throughout the present century. The comparison of the projected IDF curves for the period 2006–2099 with that obtained using GCM hindcasts for the based period (1971–2005) revealed an increase in rainfall intensity for shorter durations and a decrease for longer durations. The uncertainty in rainfall intensity for different return periods for shorter duration is found to be 2 to 6 times more compared to longer duration rainfall, which indicates that a large increase in rainfall intensity for short durations projected by GCMs is highly uncertain for peninsular Malaysia. The IDF curves developed in this study can be used for the planning of climate resilient urban water storm water management infrastructure in Peninsular Malaysia.

**Keywords:** rainfall intensity-duration-frequency curves; statistical downscaling; climate change; general circulation model; peninsular Malaysia

---

## 1. Introduction

Rainfall Intensity-Duration-Frequency (IDF) curves are one of the most frequently used tools in hydrology and water resources for the planning, design, and operation of hydraulic infrastructures [1]. The expected increase in rainfall intensity and frequency due to climate change can alter the IDF curves [2,3]. In such situations, the urban storm water management infrastructure based on IDF curves developed using the observed data can become insufficient to deal with the unexpected increase in runoff [4–6]. Several studies assessed the impact of climate change in designing urban water management infrastructure in Canada [7,8], Sweden [9], Vietnam [10], United Kingdom [11], and United States [12]. These studies have a consensus that urban water management infrastructure

will not be able to mitigate the impact of increased rainfall intensity if the design is not rectified properly considering future climate change. Estimation of adaption investment proportional to climate related risks is one of the vital challenges in infrastructure planning and management [13]. Development of an optimized and secure water resources management system is even more significant [14]. Analysis of the cost and adaption investment due to impacts of climate change is of prime importance in the planning of water management infrastructure systems [15]. Significant cost-effective steps should be taken for the identification of water resources investments that can reduce risks [16,17]. For such analyses, it is very important to quantify the impact of climate change on urban water management infrastructure.

Future climate change is often studied using the projections of General Circulation Models (GCMs). Numerous studies reported that there are several uncertainties in climate projections [18]. Incorporation of climate uncertainties in IDF curves can facilitate better decision making in urban hydraulic infrastructure planning and management. It can also be used for the assessment of climate change impacts on soil erosion with uncertainty levels. However, the projections of GCMs are not able to provide reliable information on spatial scales below about 200 km, and therefore, cannot be used for impact assessment on a local scale [19,20]. In order to assess climate change on local scales, a technique known as downscaling is used to bridge the gap between the higher resolution GCMs and the local climatic process [21,22].

Climate downscaling is broadly classified as statistical or dynamical downscaling. Due to simplicity and lower computational cost, a statistical downscaling technique is widely used for assessing climate change on local scales [23,24]. Ahmed et al. [20] reported that statistical downscaling is often preferred for its simplicity, ease of use, and flexibility, without compromising on downscaling accuracy [19,20,25]. Pour et al. [26] reported that statistical downscaling is more appropriate as it allows scenarios to be tailored for specific localities, scales, and problems. The statistical downscaling methods are broadly subdivided as perfect prognosis (PP) and model output statistics (MOS) [27]. A statistical relationship between observed climate variables (predictand) and observed large-scale predictors is established in PP, while the statistical relationship between GCM simulated predictors and observed climate variable is established in MOS. The statistical relationship is then used for the projection of climate variables using GCM simulated predictors for future scenarios. Among statistical downscaling methods, MOS is the most widely used due to its ability to explicitly account for GCM-inherent error and bias [21,26,28]. However, the relationship between GCM-simulated climate variables and the observed climate variable are often very complex, particularly in the case of rainfall. Therefore, non-parametric and non-linear methods are preferred over parametric and linear methods for correction of biases in GCM simulation [28]. Thus, distribution-wise bias correction functions are mostly widely used for the correction of biases in GCM simulations [20]. Among the distribution-wise bias correction approach, the Quantile Mapping (QM) is most widely used to correct GCM biases across the empirical cumulative distribution function [29]. The QM is a non-parametric bias correction method, and is generally applicable for all possible distributions of rainfall without any assumption on rainfall distribution. Therefore, it is widely used in recent years for the correction of biases for rainfall downscaling [30–32].

The main objective of this study is to develop IDF curves with related uncertainties under climate change scenarios. A MOS approach is used to downscale daily rainfall from eight CMIP5 (Coupled Model Intercomparison Project, Phase 5) GCMs under two radiative pathways concentration (RCP) scenarios namely, RCP 4.5 and RCP 8.5. An artificial neural network (ANN)-based model is employed for the disaggregation of projected daily rainfall to hourly maximum rainfall, which is then used for the development of IDF curves. In addition, model correction factors (MCF) are used to overcome over- and under-estimation in the projected IDF curves. Finally, the 1st quartiles, medians, and 3rd quartiles of projected rainfall intensities are estimated for developing IDF curves with uncertainty levels.

## 2. Study Area and Datasets

### 2.1. Study Area

The methodology adopted in this study is implemented across some stations of Peninsular Malaysia. Locations of the selected stations with station IDs are given in Figure 1. Geographically, Peninsular Malaysia is situated in the tropics between latitude 1.20° north to latitude 6.40° north, and longitude 99.35° east to longitude 104.20° east. The mean temperature ranges from 21 °C to 32 °C [33,34]. The climate of the study area is categorized by the two regimes of monsoon winds, i.e., northeast monsoon and southwest monsoon. The southwest monsoon regime exists during May to August, and the northeast monsoon regime during November to February. The whole country has a drier period during the southwest monsoon, whereas the northeast monsoon brings heavy rainfall in the coastal areas of peninsular Malaysia. In contrast, the areas sheltered under mountainous topography are almost free from their influence. Furthermore, maximum rainfall is recorded during the transition period between monsoon regimes, the 'inter-monsoon period' (March–April and September–October), especially at the rainfall stations located in the western areas [35,36].

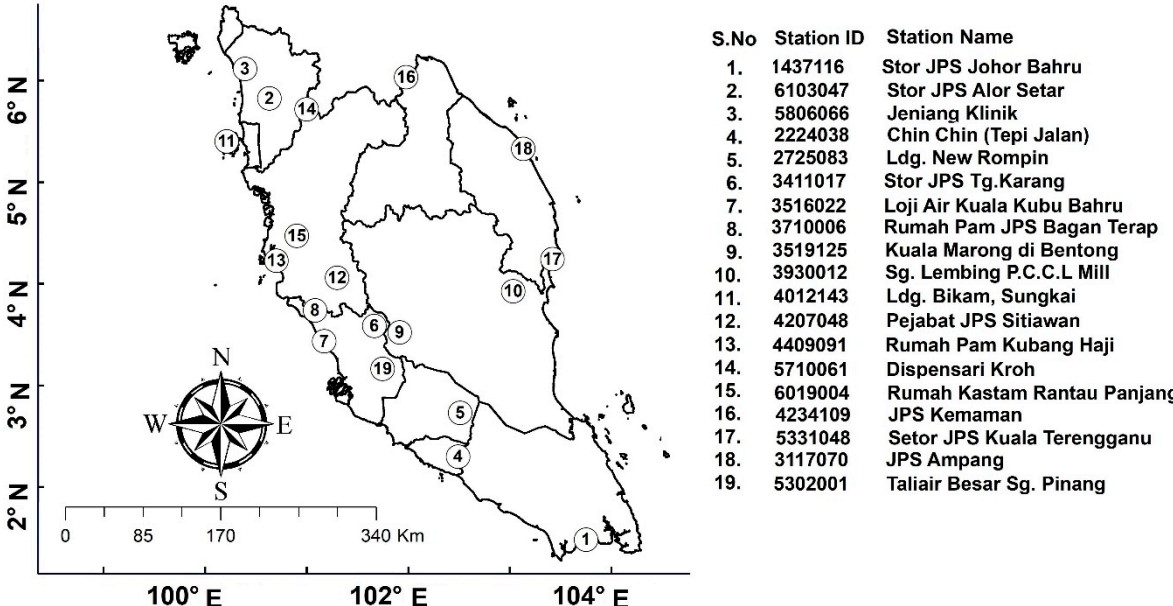

**Figure 1.** Location of the selected rain gauges on the map of peninsular Malaysia. The number in circle represents the station name and ID, as shown on the right side in figure.

### 2.2. Data and Sources

Thirty-five years (1971–2005) of hourly rainfall data—recorded at 19 stations mostly distributed over urban areas of Peninsular Malaysia, collected from the Department of Irrigation and Drainage (DID), Malaysia—is used in this study. The GCM simulations of CMIP5 that form the basis for the fifth assessment report (AR5) of the Intergovernmental Panel on Climate Change (IPCC) are used.

It is not feasible to use all the CMIP5 GCMs for climate change projection and impact assessment due to constraints in human and computational resources [37]. In practice, a small ensemble of GCMs is selected, considering that they will able to provide whole range of uncertainties in the projections [26]. In this study, one GCM from each participating modeling center in CMIP5 that have future projections for both RCP 4.5 and RCP 8.5 are selected, considering that projections from all centers will provide the full range of uncertainty in future projections. Thus, total 8 GCMs are selected in this study, as listed in Table 1.

**Table 1.** List of IPCC CMIP5 GCMs used in the present study.

| Centre(s) | Model | Resolution (Lat × Long) |
|---|---|---|
| Beijing Climate Center China | BCC-CSM1.1 | 2.8° × 2.8° |
| Commonwealth Scientific and Industrial Research Organization/Queensland Climate Change Centre of Excellence Australia | CSIRO-Mk3.6.0 | 1.8° × 1.8° |
| Institut Pierre Simon Laplace France | IPSL-CM5A-MR | 1.25° × 2.5° |
| Atmosphere and Ocean Research Institute (The University of Tokyo), National Institute for Environmental Studies, and Japan Agency for Marine-Earth Science and Technology, Japan | MIROC-ESM | 2.8° × 2.8° |
| Met Office Hadley Centre UK | HadGEM2-ES | 1.25° × 1.875° |
| Meteorological Research Institute Japan | MRI-CGCM3 | 1.12° × 1.125° |
| National Center for Atmospheric Research USA | CCSM4 | 0.94° ×1.25° |
| Bjerknes Centre for Climate Research, Norwegian Meteorological Institute, Norway | NorESM1-M | 1.90° × 2.5° |

The simulated historical and future daily precipitation of eight GCMs over the climatic domain of peninsular Malaysia is collected from IPCC portal (http://www.ipccdata.org/sim/gcm_monthly/AR5/ReferenceArchive.html). The GCMs simulated historical rainfall for the period of 1971–2005 and projected rainfall for 2006–2099 under two RCPs (RCP 4.5 and RCP 8.5) is used in this study. The RCP 4.5 is an intermediate pathway scenario which provides a common platform for climate models to explore the climate system response to stabilizing the anthropogenic components of radiative forcing [38]. The latest policy of the global community is environmental sustainability and lower greenhouse gas emissions, and therefore, the RCP 4.5 scenario is often considered to be a very-good-case scenario in the context of recent policy directions [39]. On the other hand, RCP 8.5 provides data which most closely resemble the present observations so far, and therefore, gives the possible highest impact. For cost-effective risk analysis, knowledge of the possible range of impacts is very important. As RCP 4.5 and RCP 8.5 provide the best- and worse-case scenarios, respectively, those are selected in this study.

## 3. Methodology

### 3.1. Procedure

The methodology adopted in this study for updating IDF curves under climate change scenarios is represented by the flow chart shown in Figure 2.

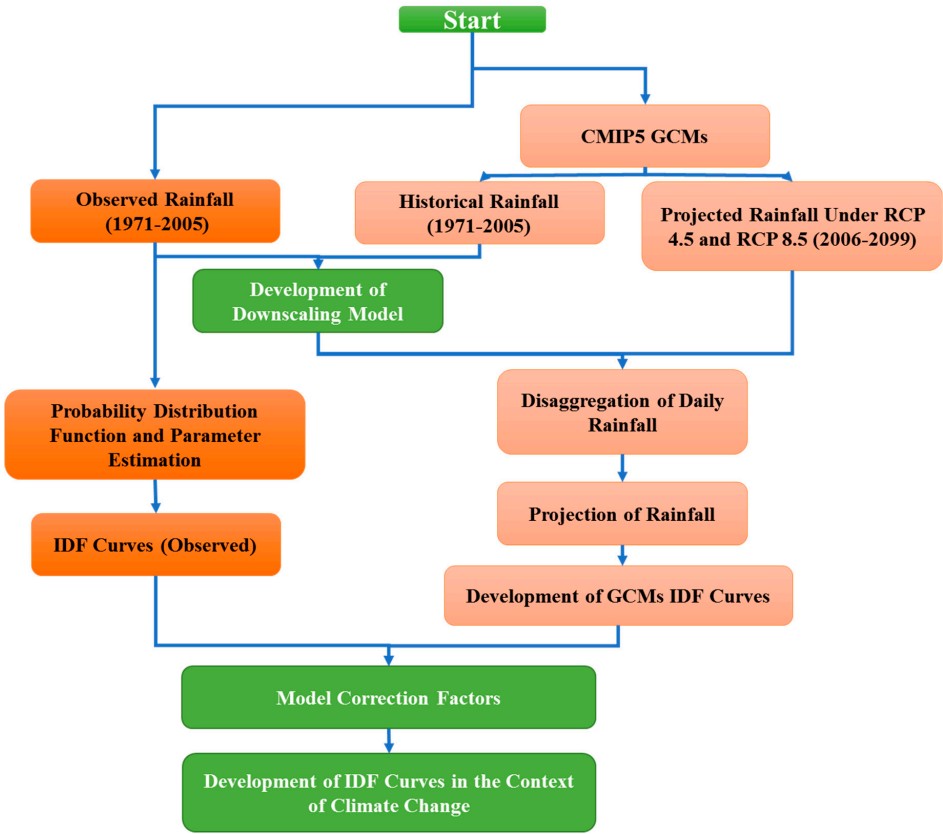

**Figure 2.** Flow Chart of the Methodology.

The procedure is outlined below:

1.　The GCM simulations for historical (1971–2010) and future (2006–2099) periods are interpolated to observed locations.
2.　MOS downscaling model is developed where quantile mapping (QM) is used to derive the bias correction factors by comparing GCM simulated rainfall with observed rainfall for the period (1971–2005).
3.　The bias correction factors derived from the historical period (1971–2005) are then applied on simulated GCM rainfall for different RCPs for the period 2006–2099.
4.　An artificial neural network (ANN) model is developed to disaggregate daily rainfall data to hourly rainfall data. The model is calibrated and validated with observe hourly rainfall data for the period 1971–2005.
5.　The ANN disaggregation model is used to generate hourly rainfall data from daily rainfall projected for the period 2006–2099.
6.　IDF curves are generated by fitting observed annual maximum of hourly rainfall data (1971–2005) with most suitable probability density function (PDF) and parameter estimation method.
7.　The disaggregated rainfall data are used to generate time series of annual maximum of hourly rainfall to develop IDF curves for climate change scenarios.
8.　The model correction factors (MCFs) are estimated for all the durations of rainfall by fitting the average of the ratios of the projected return periods to observed return periods in a polynomial equation.
9.　The MCFs are applied on the return periods of rainfall durations for future period to generate the IDF curves. The IDF curves are generated for all the 8 GCMs for both the RCP 4.5 and RCP 8.5, separately.
10.　Finally, the IDF curves are developed with uncertainty level, by estimating the 1st quartile, median and 3rd quartile of the return periods of different rainfall durations obtained from IDF curves generated for eight GCMs.

The methods used in this study are discussed below.

### 3.2. Selection of Appropriate Probability Density Function and Paramter Estmation Method

The PDF used for fitting annual maximum of hourly rainfall significantly influences the shape of IDF curves. Furthermore, estimated values of PDF parameters vary significantly with parameter estimation methods, and thereby, influence the nature of IDF curves. Therefore, the choice of an appropriate PDF and parameter estimation method is very important. Various PDFs are used for fitting annual maximum of hourly rainfall data, but there is no general criterion for the selection of PDF for frequency analysis of extreme rainfall events. A PDF selected for a location may not exhibit good results at another location. Therefore, comparing various PDFs for the selection of the most appropriate one is considered good practice [40]. In this study, four of the most commonly used PDFs namely, Generalized Pareto (GP), Gumbel, Generalized Extreme Value (GEV), and Exponential, and four commonly used parameter estimation methods, namely Generalized Maximum likelihood (GMLE), L-moments, Maximum likelihood (MLE), and Bayesian, are compared based on negative log likelihood goodness of fit tests. The negative likelihood ratio provides in-built strength for a test which will rule in or out probabilities; therefore, it is widely used for assessing the performance of a diagnostic test [40]. The major purpose of comparison is to find the best PDF and parameter estimation method over the selected stations. Finally, the most suitable PDF with the best parameter estimation method is used for developing the IDF curves from hourly time series for both projected and historical rainfall. These PDF are briefly explained in Table 2.

**Table 2.** The PDFs compared for selection of the best PDF.

| Functions | Equations | Parameters |
|:---:|:---:|:---|
| GEV | $f(x) = \begin{cases} \frac{1}{\sigma} exp(-(1+kz)^{-1/k})(1+kz)^{-1-1/k} & k \neq 0 \\ \frac{1}{\sigma} \exp{(-z - \exp{(-z)})} & k = 0 \end{cases}$ | where, $z = \frac{x-\mu}{\sigma}$ |
| Exponential | $f(x) = \begin{cases} \lambda exp(-\lambda x) & x \geq 0 \\ 0 & x < 0 \end{cases}$ | and $k$ = shape parameter |
| GP | $f(x) = \begin{cases} \frac{1}{\sigma}(1+kz)^{-1-1/k} & k \neq 0 \\ \frac{1}{\sigma} exp(-z) & k = 0 \end{cases}$ | $\mu$ = location parameter $\sigma$ = scale parameter |
| Gumbel | $f(x) = \frac{1}{\sigma} exp(-z - exp(-z))$ | |

### 3.3. Rainfall Downscaling and Projections

The following procedure is used for downscaling and projection of daily rainfall at observed locations under RCP scenarios:

1. The GCM simulated rainfall is interpolated at each station using inverse an weighting distance method to generate GCM simulations at the observed location.
2. The QM is used to compute the biases in GCMs by comparing 70% of the randomly-selected observed and GCM simulated daily rainfall for the period 1971–2005. The QM bias correction parameters are validated with the remaining 30% of observed and GCM simulated daily rainfall for the period 1971–2005.
3. The derived QM parameters are used to correct the biases in the simulated daily rainfall of GCMs for both the RCP 4.5 and RCP 8.5 for the period 2006–2099.

The performance of downscaling model is evaluated using mean absolute error (MAE), normalized root-mean-square error (NRMSE), Percent Bias (PBIAS), coefficient of determination ($R^2$), and Nash–Sutcliffe Coefficient of Efficiency (NSE),

$$MAE = \frac{1}{n} \sum_{i=1}^{n} |x_{obs,i} - x_{sim,i}| \qquad (1)$$

$$NRMSE = \frac{\left[\frac{1}{n}\sum_{i=1}^{n}\left(x_{sim,i} - x_{obs,i}\right)^2\right]^{\frac{1}{2}}}{\overline{x}_{obs,i}} \tag{2}$$

$$PBIAS = 100 * \frac{\sum_{i=1}^{n}\left(x_{sim,i} - x_{obs,i}\right)}{\sum_{i=1}^{n} x_{obs,i}} \tag{3}$$

$$R^2 = \frac{\sum_{1}^{n}\left(x_{obs,i} - \overline{x}_{obs}\right)\left(x_{sim,i} - \overline{x}_{sim}\right)}{\sqrt{\sum_{i=1}^{n}\left(x_{sim,i} - \overline{x}_{sim}\right)^2 \sum_{i=1}^{n}\left(x_{obs,i} - \overline{x}_{obs}\right)^2}} \tag{4}$$

$$NSE = 1 - \frac{\sum_{i=1}^{n}\left(x_{sim,i} - x_{obs,i}\right)^2}{\sum_{i=1}^{n}\left(x_{obs,i} - \overline{x}_{obs,i}\right)^2} \tag{5}$$

where, $x_{sim,i}$ and $x_{obs,i}$ are the $i$th simulated and observed data, and $n$ is the number of the observations.

### 3.4. Disaggregation of Daily Rainfall and Generation of Projected IDF Curves

Different statistical and data-driven models have been used for temporal disaggregation of rainfall. A number of recent studies reported promising performance of ANN in temporal disaggregation of rainfall [41–47]. Zhang et al. [43] used ANN for disaggregation of rainfall for West-Central Florida, and reported that the disaggregation of rainfall using ANN is promising. Mirhosseini et al. [44] used ANN for the disaggregation of precipitation data simulated by five combinations of global and regional climate models. They compared the results with disaggregated rainfall derived using a stochastic method, and showed that the ANN model provides superior performance in estimating maximum rainfall depths. They showed that the IDF curves developed for future rainfall intensities was independent of the temporal disaggregation method used. Mirhosseini et al. [46] compared the performance of ANN and stochastic rainfall disaggregation methods for the development of IDF curves, and reported that the results of the both methods were in agreement with the observed precipitation pattern.

The ANN-based disaggregation technique is used in this study for the disaggregation of the daily precipitation data to hourly rainfall. The ANN-based rainfall disaggregation method proposed by [41] is used in this study. Burian et al. [41] used a three-input and four-output ANN model where three hourly rainfall amounts (rainfall of preceding, current and successive hours) were used as input to generate four consecutive 15-min rainfall amounts. This concept was used by others for the disaggregation of daily rainfall [42,48]. In recent years, Kim et al. [45] used a single input and 11 outputs ANN model for the spatial aggregation of areal rainfall. Kim et al. [48] used a single input and 12 outputs ANN model for the disaggregation of areal rainfall. In the present study, ANN models are developed to disaggregate hourly rainfall data in two stages. In the first step, a three- and four-output ANN model is developed to disaggregate daily rainfall to six-hour rainfall pattern. Rainfall of three consecutive days (preceding, current and successive) is used as input to generate the four six-hour rainfall amount of the current day as in [42]. In the next stage, a three- and six-output ANN model is employed to disaggregate the six-hour rainfall amount to the 1-hr rainfall pattern. Consecutive three six-hour rainfall amounts (preceding, current and successive) are used as inputs to generate the hourly rainfall of the current six hours. Hourly observed rainfall data is aggregated to generate six-hour and one-day rainfall amounts for the training and validation of the disaggregation models.

A resilient backpropagation with a weight backtracking method is used to train the neural networks. In this study, the disaggregation model is trained with the observed daily rainfall data as input and hourly rainfall as output. The observed hourly rainfall is repeatedly compared with the predicted output, and then the corresponding error is measured. The error is applied to adjust the weights and biases to efficiently disaggregate the daily rainfall to hourly rainfall. There is no rule to decide the number of hidden layers and neurons in ANN. In this study, an optimization method is used where optimum structure of ANN is selected based on a trial and error method. Different ANN models are developed with optimum structures for the disaggregation of daily rainfall data at

different stations. The ANN models were calibrated and validated with observed hourly rainfall data for the period 1971–2005. About 70% of the observed data (January 1971–June 1995) is used for model calibration, while the remaining 30% (July 1995–December 2005) is used for model validation.

The calibrated and validated disaggregation models are then used to disaggregate the daily rainfall data projected by the GCMs for scenarios, RCP 4.5 and RCP 8.5. The disaggregated hourly rainfall data are then used for the development of IDF curves for the projected climate. The IDF curves are prepared for all the GCMs under two scenarios, RCP 4.5 and RCP 8.5. Finally, the IDF curves prepared for future climate change scenarios are compared with the observed IDF curves. The changes in rainfall intensity and duration for various return periods of rainfall events projected by different GCMs is analyzed to assess the effects of climate change on rainfall IDF in peninsular Malaysia.

*3.5. Model Correction Factors*

The observed IDF curve needs to be updated with respect to the expected changes in rainfall due to climate change. IDF estimates based on rainfall intensity are biased by the assumptions that the annual maximum of rainfall for a particular duration occurs in one of the non-overlapping intervals (such as maximum 1 h rainfall occurs in a non-overlapping one-hour interval). It also considers that the rainfall amounts in different durations are independent. Therefore, disaggregated data shows significant underestimation or overestimation as compared to the observed data for various rainfall durations. The classical way to eliminate this bias is to use a correction factor [49–51]. The correction factors correct the bias in disaggregated rainfall intensities using the differences and ratios between observed and disaggregated intensities for different rainfall durations which has been described details in [51].

To develop the MCFs, first of all, the ratios of modeled-(GCM simulated) to-observed (gauged) rainfall intensity of all the return periods (2, 5, 10, 25, 50 and 100 years) are computed separately for a specific duration (e.g., 1 h). Then, the average of the ratios of return periods is computed for that specific duration (e.g., 1 h). In the same way, the average of the ratios of the return periods is computed separately for all others durations. Finally, to estimate the MCFs, the average of the ratios of the return periods obtained for the all the durations of rainfall are fitted into their polynomial equation. Due to non-linear relationship between average ratio of the return period and the rainfall duration, a polynomial equation is used for the fitting of data and estimation of MCF [51]. The MCFs are developed for all the durations of rainfall projected by a GCM. The general equation for developing MCFs would be:

$$y = a\, x^2 + b\, x + c \tag{6}$$

where $y$ is the MCF and $x$ is the average of the ratios of all the return periods for a specific duration of rainfall projected by a GCM. The MCF developed for a specific duration of GCM is then multiplied with the all the observed returns periods 2, 5, 10, 25, 50, and 100 years (obtained from gauged rainfall data) for the same duration of rainfall (i.e., the MCFs developed for 1 h duration of rainfall is multiplied with observed return periods for 1 h duration). The return periods thus produced are representative of the IDF curve within the context of climate change.

For a better understanding of the procedure used for estimating MCFs, an example is provided here. Suppose, for BCC-CSM 1.1 projected 1 h rainfall duration, the ratios of rainfall intensity of modelled (GCM simulated) to observed rainfall of 2, 5, 10, 25, 50, and 100 years return periods are $r_1$, $r_2$, $r_3$, $r_4$, $r_5$, and $r_6$ respectively (see Table 6 in Section 4.5). The average of the ratios of all these six return periods (2, 5, 10, 25, 50 and 100 years) for 1 h duration rainfall will be, $x_1 = (r_1 + r_2 + r_3 + r_4 + r_5 + r_6)/6$. In the same way, $x_2$, $x_3$, $x_4$, $x_5$, $x_6$, and $x_7$ are computed for the durations 3, 6, 12, 24, 48, and 72 h respectively (see Table 6 (last line) and Table 7 in Section 4.5 the ratios for model BCC-CSM1.1). To calculate the MCFs ($y_1$, $y_2$, $y_3$, $y_4$, $y_5$, $y_6$, $y_7$) for all these durations of rainfall projected by BCC.CSM1.1, the average of the ratios of the return periods obtained for all the durations are put into their polynomial equation (see Figure 8 and Equation (7) in Section 4.5). In this way MCFs are obtained separately for all durations (Table 9 in Section 4.5). Finally, the MCF developed for a GCM (BCC-CSM1.1) is multiplied with the observed (gauged) return periods (i.e., 2, 5, 10, 25, 50, and

100 years) for the same duration of rainfall. For example, the MCF of 1 h duration is multiplied with the observed 1 h duration rainfall for return periods, 2, 5, 10, 25, 50, and 100 years.

Following the procedure discussed above, the IDF curves are developed for all the GCMs under two climate change scenarios, i.e., RCP 4.5 and RCP 8.5. To assess the uncertainty, the projected IDF curves are estimated separately for each return period at the 1st quartile, median, and 3rd quartile, as well and the maximum and the minimum values of all the eight GCMs.

## 4. Results and Discussion

### 4.1. Determination of Probability Density Function & Parameter Estimation Method

Log-likelihood estimations at a station located in Kedah (station ID 5806066) for the PDFs and parameter estimation methods used are shown in Table 3 as an example. The lowest log-likelihood estimates at this station are found for GEV with the MLE parameter estimation method. The PDFs and parameter estimation methods are compared for all the rainfall maxima (1 to 72 h of rainfall) at all the 19 rainfall stations. The results for the best PDF with parameter estimation methods compared for the stations used are represented by alphabets in Table 4. The GEV is found to be the most suitable PDF and MLE as the best parameter estimation method at most of the stations of peninsular Malaysia used in this study. Other distributions are found to be suitable only in a few locations. Therefore, GEV parameters estimated using MLE is used for the development of IDF curves using observed rainfall at different locations of peninsular Malaysia.

**Table 3.** The results of goodness-of-fit test at a station located in Kedah.

| Estimators | Functions (PDFs) | Durations (Hour) | | | | | | |
|---|---|---|---|---|---|---|---|---|
| | | 1 h | 3 h | 6 h | 12 h | 24 h | 48 h | 72 h |
| MLE | GEV | 294.88 | 302.74 | 302.46 | 307.9 | 319.75 | 333.06 | 341.18 |
| | Gumbel | 295.05 | 302.81 | 302.77 | 308.17 | 319.79 | 333.12 | 341.2 |
| | Exp | 331.19 | 348.82 | 352.79 | 355.99 | 362.27 | 372.3 | 379.05 |
| | GP | 324.23 | 335.72 | 348.73 | 350.64 | 348.4 | 380.8 | 378.85 |
| GMLE | GEV | 296.32 | 303.97 | 302.84 | 308.29 | 320.88 | 335.53 | 342.74 |
| | Gumbel | 295.05 | 302.81 | 302.77 | 308.17 | 319.79 | 333.12 | 341.2 |
| | Exp | 331.19 | 348.82 | 352.79 | 355.99 | 362.27 | 372.3 | 379.05 |
| | GP | 445.27 | 493.66 | 505.88 | 504.24 | 498.67 | 504.61 | 504.26 |
| L-moments | GEV | 296.32 | 303.97 | 302.84 | 308.29 | 320.88 | 335.53 | 342.74 |
| | Gumbel | ∞ | ∞ | ∞ | ∞ | ∞ | ∞ | ∞ |
| | Exp | ∞ | ∞ | ∞ | ∞ | ∞ | ∞ | ∞ |
| | GP | 445.27 | 493.66 | 505.88 | 504.24 | 498.67 | 504.61 | 504.26 |
| Bayesian | GEV | 5395.3 | 6183.1 | 6027.8 | 6743.8 | 9059.1 | 12867 | 14697 |
| | Gumbel | ∞ | ∞ | ∞ | ∞ | ∞ | ∞ | ∞ |
| | Exp | ∞ | ∞ | ∞ | ∞ | ∞ | ∞ | ∞ |
| | GP | ∞ | ∞ | ∞ | ∞ | ∞ | ∞ | ∞ |

Exp = Exponential; MLE = Maximum Likelihood Estimation; GMLE = Generalized Extreme Estimation; GP = Generalized Pareto.

**Table 4.** Best PDF and parameter estimation methods over selected stations.

| Station ID | State | 1 h | 3 h | 6 h | 12 h | 24 h | 72 h |
|---|---|---|---|---|---|---|---|
| 1437116 | Johor Bahru | A | A | A | B | B | B |
| 5806066 | Kedah | A | A | A | A | A | A |
| 6103047 | Kedah | A | A | A | A | A | A |
| 2224038 | Melaka | A | A | A | A | B | B |
| 2725083 | Niger Sembilan | A | A | A | A | A | A |
| 3516022 | Selangor | A | A | A | A | A | A |
| 3411017 | Selangor | B | B | B | B | A | A |

**Table 4.** *Cont.*

| Station ID | State | 1 h | 3 h | 6 h | 12 h | 24 h | 72 h |
|---|---|---|---|---|---|---|---|
| 3710006 | Selangor | A | A | A | A | A | A |
| 3519125 | Pahang | A | A | A | A | A | A |
| 3930012 | Pahang | A | A | A | A | A | A |
| 5302001 | Pinang | B | C | B | C | C | B |
| 4012143 | Perak | A | A | A | A | A | A |
| 4207048 | Perak | A | A | A | A | A | A |
| 5710061 | Perak | A | A | A | A | E | E |
| 4409091 | Perak | D | D | D | D | D | F |
| 6019004 | Kelantan | D | D | F | F | F | D |
| 4234109 | Terengganu | A | A | A | A | A | A |
| 5331048 | Terengganu | A | A | A | A | A | A |
| 3117070 | W. Persekutuan | A | A | D | D | D | A |

A = (GEV MLE) B = (GEV GMLE) C = (GEV LM) D = (GP MLE) E = (GUMBLE MLE) and F = (EXPONETIAL MLE).

### 4.2. The IDF Curves based on Historical Rainfall

The hourly rainfall data for the period 1971–2005 are used to generate the IDF curves for base years. These IDF curves are generated for the rainfall durations of 1 to 72 h and the return periods of 2, 5, 10, 25, 50, and 100 years for all the stations. The IDF curves from observed data at a station located Kedah are shown in Figure 3 as an example. It is observed that the intensity of rainfall is low for shorter return periods, and it increases gradually for higher return periods, while the rainfall intensity decreases in descending order for higher duration.

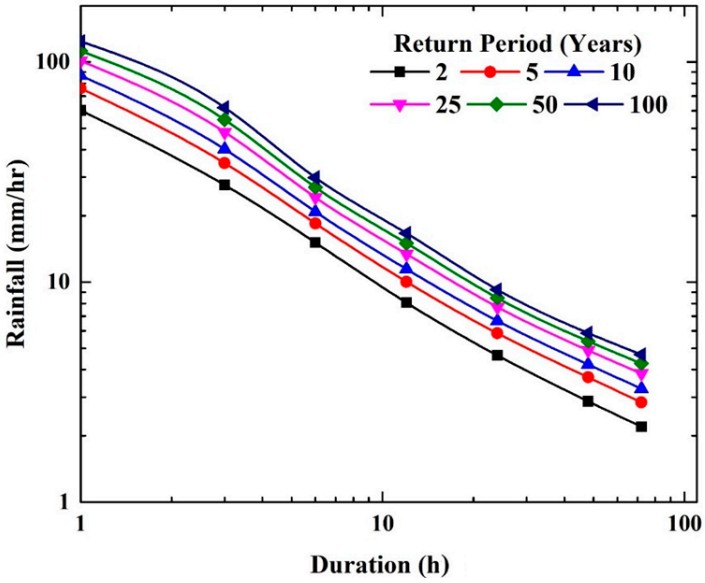

**Figure 3.** Historical IDF curves developed using GEV-MLE for observed (gauged) rainfall data (1971–2005) at a station located in Kedah.

### 4.3. Climate Downscaling and Projections

QM-based MOS downscaling models are constructed through the calibration and validation processes [52]. Downscaling models are developed for each GCM and rain gauge stations separately. The downscaling models are developed at each station for each calendar month. Therefore, total 1824 models [12 (month) × 19 (station) × 8 (CMIP5 GCMs) = 1824] are calibrated and validated for the downscaling and projection of rainfall in the study area. The QM parameters are derived using 70% of daily rainfall data selected randomly for the period 1971–2005. The performance of QM in bias correction is then validated with 30% of daily rainfall data. It should be noted that GCMs are used

for simulations of climate, and therefore, day to day exact simulation of daily rainfall is not expected using GCMs. Therefore, the matching of daily observed and GCM simulated data is not practical. However, GCMs are expected to simulate the seasonal variability and overall climate of a region. Therefore, daily bias corrected data are usually converted to monthly data to show the performance of downscaling model. Observed and downscaled rainfall for the period 1971–2005 is compared to assess the performance of the downscaling model. A comparison of monthly downscaled and observed rainfall for all GCMs at station Kedah is presented in Figure 4. As the data for estimation and validation of QM bias correction parameters were randomly selected over the period 1971–2005, the observed and downscaled rainfall data for both periods are presented together in the figure. The performance of MOS downscaling model is evaluated using the statistical indices MAE, NRMSE, PBIAS, $R^2$, and NSE. The calibration and validation values of these indices for this station are presented in Table 5. It is found that all the GCMs performed well in term of all statistical indices used. The MAE values are in the range of 0.15–0.56 and the NRMSE value are in the range of 6.1–21.3. The PBIAS values are between 0.1 and 4.1. The values very near to zero indicate good performance of the models. The $R^2$ values are always found very near to 1, and NSE is above 0.9 in most of the cases. Similar types of results are also found in other stations for all models. The statistical indices values are very close to each other for different GCMs. Hence, it can be concluded that the QM-based MOS downscaling model has the capability of downscaling daily rainfall in the study area. The calibrated and validated MOS models are used for the projection of rainfall under RCP scenarios. Using the MOS downscaling model, rainfall is projected for the period of 2006–2099 for RCP 4.5 and RCP 8.5.

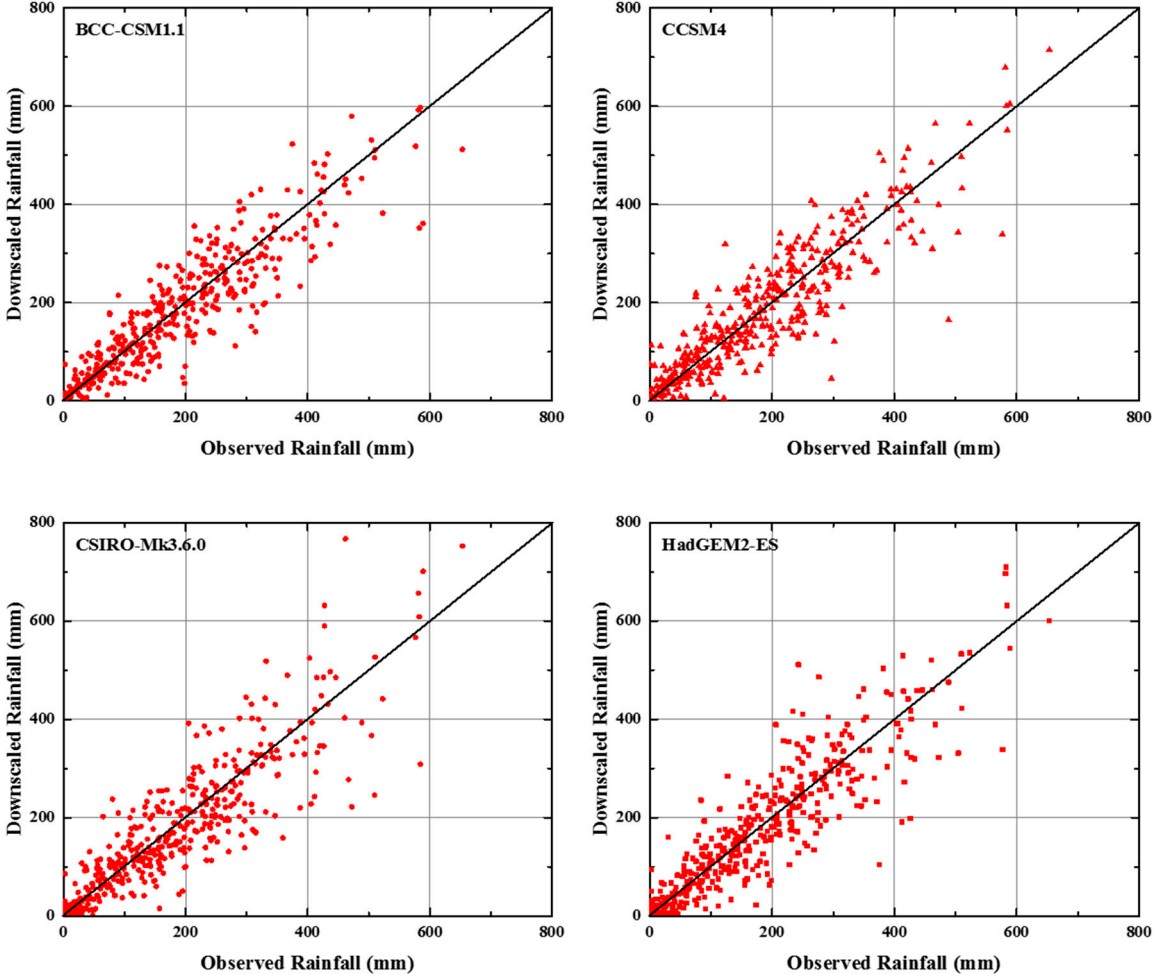

**Figure 4.** *Cont.*

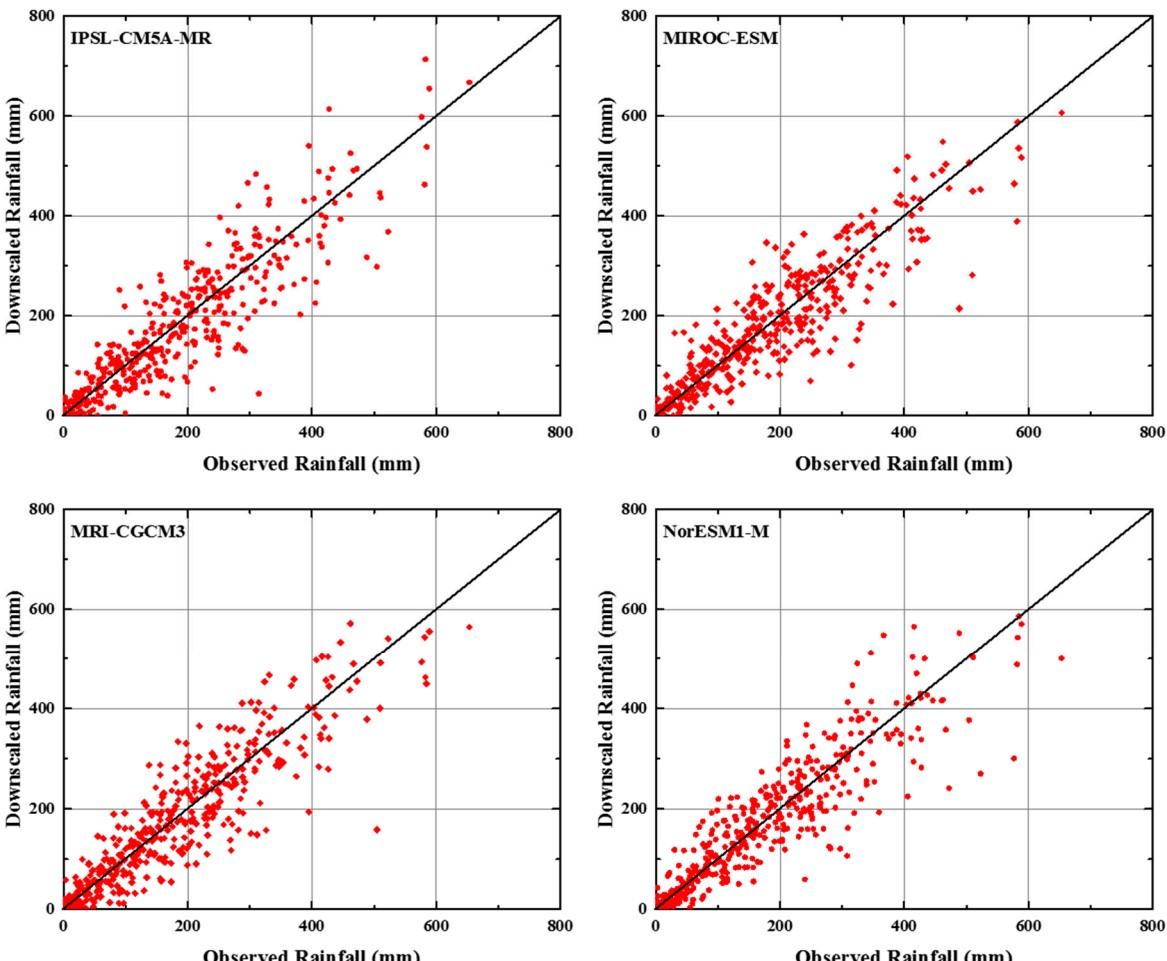

**Figure 4.** Downscaled and observed rainfalls for various GCMs at station Kedah.

**Table 5.** Calibration and validation of downscaling model for various GCMs at station Kedah.

| Station ID | Indices | Model | MAE | NRMSE % | PBIAS % | NSE | $R^2$ |
|---|---|---|---|---|---|---|---|
| | | BCC-CSM1.1 | 0.31 | 12.1 | 1.1 | 0.95 | 0.91 |
| | | CCSM4 | 0.27 | 12 | 0.6 | 0.97 | 0.93 |
| | | CSIRO-Mk3.6.0 | 0.34 | 13.1 | 2.1 | 0.94 | 0.95 |
| | Calibration | HadGEM2-ES | 0.36 | 14.6 | 0.3 | 0.94 | 0.93 |
| | | IPSL-CM5A-MR | 0.33 | 12.1 | 3.1 | 0.92 | 0.92 |
| | | MIROC-ESM | 0.21 | 8.8 | 0.2 | 0.91 | 0.96 |
| | | MRI-CGCM3 | 0.34 | 12.9 | 1.3 | 0.94 | 0.95 |
| Kedah 5806066 | | NorESM1-M | 0.15 | 6.1 | 0.5 | 0.91 | 0.93 |
| | | BCC-CSM1.1 | 0.45 | 20.6 | 2.1 | 0.93 | 0.92 |
| | | CCSM4 | 0.54 | 21.3 | 1.9 | 0.95 | 0.90 |
| | | CSIRO-Mk3.6.0 | 0.56 | 21 | 0.1 | 0.96 | 0.91 |
| | Validation | HadGEM2-ES | 0.4 | 17.2 | 4.1 | 0.94 | 0.94 |
| | | IPSL-CM5A-MR | 0.42 | 19 | 1.3 | 0.93 | 0.93 |
| | | MIROC-ESM | 0.42 | 17.9 | 1.1 | 0.93 | 0.95 |
| | | MRI-CGCM3 | 0.42 | 19.2 | 1 | 0.92 | 0.93 |
| | | NorESM1-M | 0.45 | 17.8 | 1.9 | 0.94 | 0.93 |

### 4.4. Disaggregation of Rainfall

To assess the performance of the disaggregation model, annual observed rainfall maxima and the disaggregated annual observed rainfall maxima are compared for the period of 1971–2005. The results obtained at station Kedah are shown in Figure 5 as an example. It shows a good match between observed and disaggregated rainfall maxima for different hours. Even the high rainfall values are reliably replicated by the disaggregation model. Due to space limitations, the results of only one duration for four GCMs are shown. The results of the GCMs (a) BCC-CSM 1.1 for 1 h duration of rainfall (b) HadGEM2-ES for 3 h duration of rainfall (c) Nor-ESM-M for 12 h duration of rainfall and (d) CCSM4 for 72 h duration of rainfall are presented. A similar type of results is obtained for other durations at all the stations for all the GCMs used. This indicates the efficacy of ANN-based disaggregation method in generating hourly rainfall maxima from daily rainfall data. The ANN-based disaggregation model is then used to disaggregate the daily rainfall simulations of the GCMs to generate hourly time series. This hourly time series obtained from GCMs daily rainfall is then used for development IDF curves for both historical and future simulations of GCMs. The IDF curves developed from the GCMs future simulations are the projected IDF curves under changing climate scenarios.

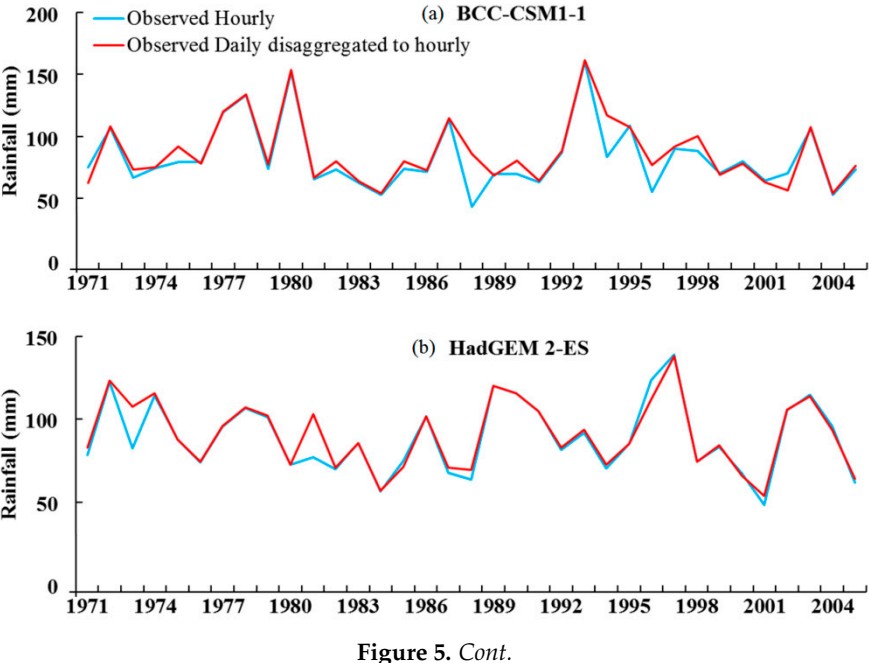

**Figure 5.** *Cont.*

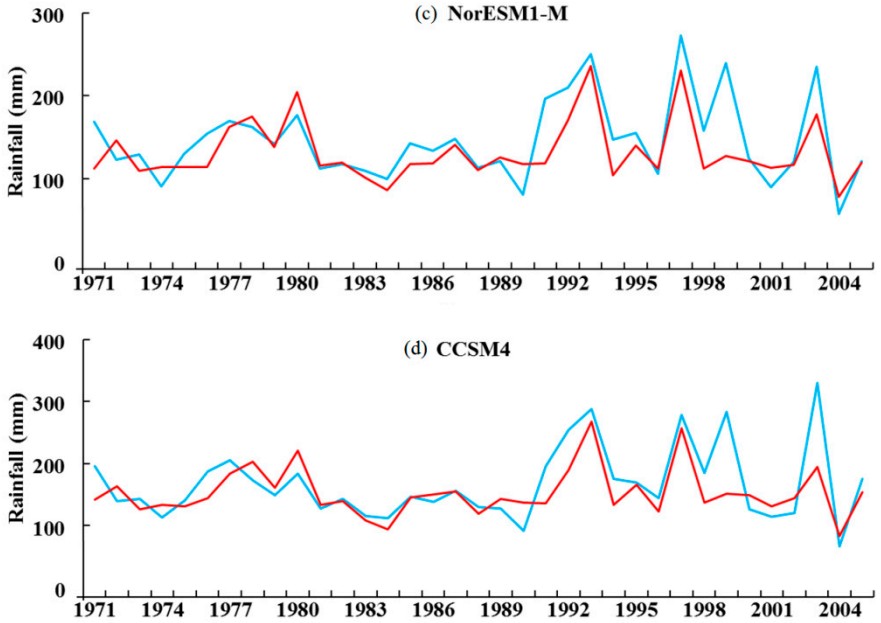

**Figure 5.** Comparison of observed hourly maximum rainfall and observed daily rainfall disaggregated to hourly maximum rainfall using ANN approach at station Kedah for (**a**) BCC-CSM 1.1 for 1 h (**b**) HadGEM2-ES for 3 h (**c**) Nor-ESM-M for 12 h and (**d**) CCSM4 for 72 h.

### 4.5. Development of IDF Curves under Climate Change Scenarios

The disaggregated hourly rainfall obtained from historical simulated and projected daily rainfall by different GCMs is used for the generation of IDF curves. The projected IDF curves are generated for all the GCMs for RCP 4.5 and RCP 8.5. Using GEV distribution, the GCM simulated historical and projected IDF curves are generated for 2, 5 10, 25, 50, and 100 years return periods. The GCMs simulated historical IDF curves and projected IDF curves for RCP 4.5 for station Kedah are shown in Figures 6 and 7 respectively. By comparing the GCM simulated historical IDF curves (Figure 6) with the IDF curves developed from observed rainfall (Figure 3), it is found that the GCMs simulations over- or under-estimates the rainfall predictions in some cases. Therefore, to overcome the problem of over or under estimation, the projected IDF curves are corrected using model correction factor.

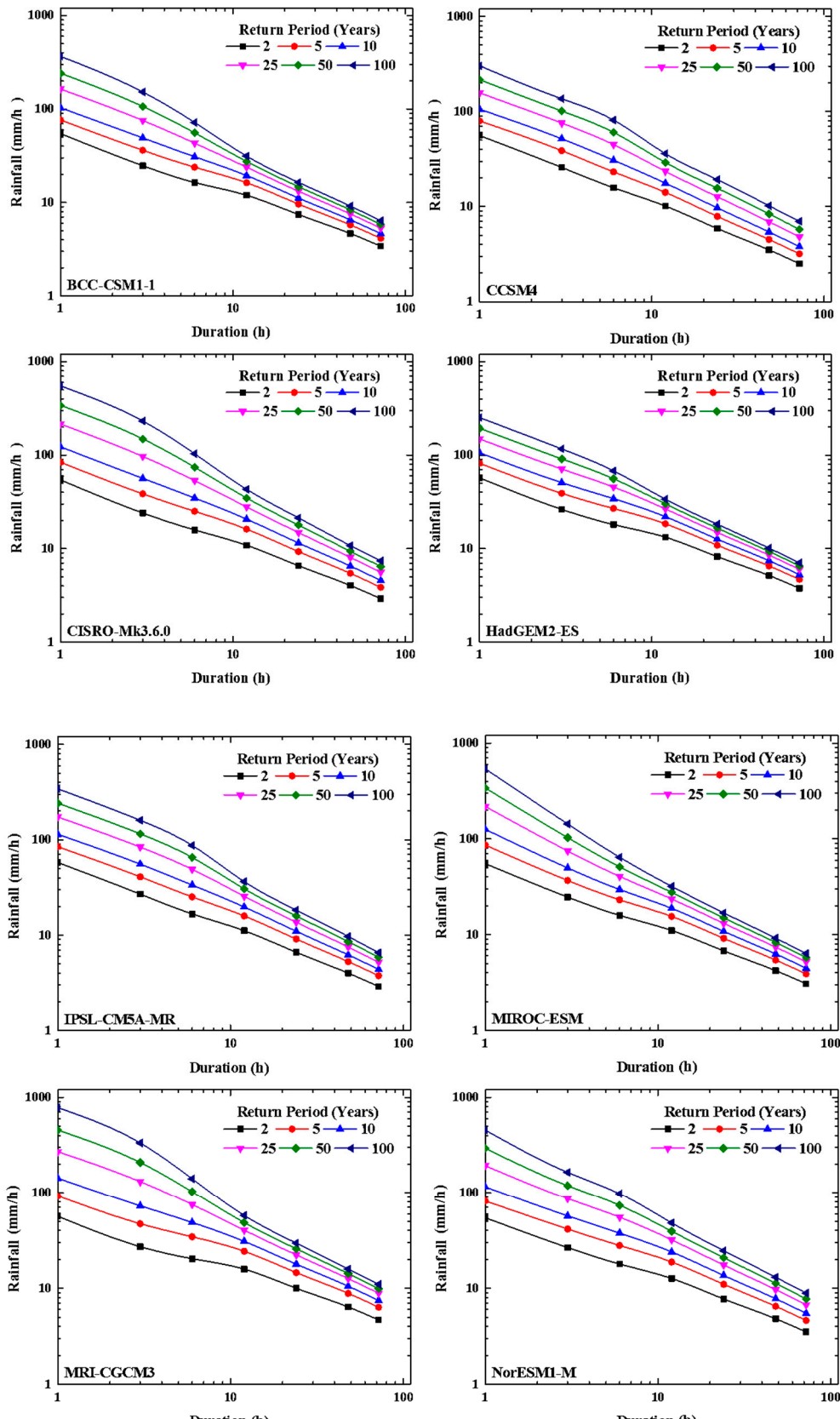

**Figure 6.** The IDF curves developed for station Kedah using GCMs simulated historical rainfall data (1971–2005).

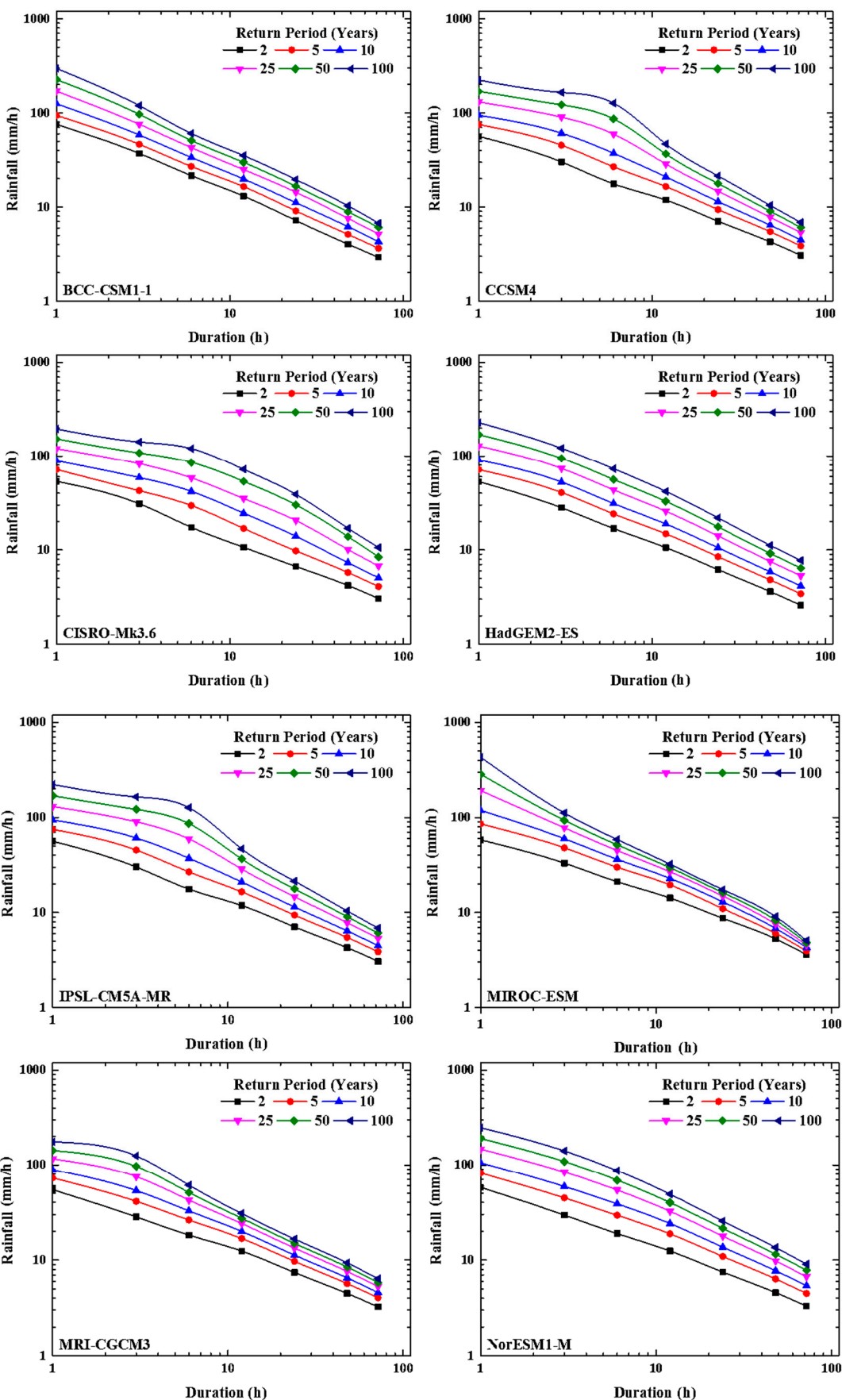

**Figure 7.** The IDF curves developed for station Kedah using GCMs projected rainfall data (2006–2099).

Model Correction Factor

The model correction factors [51] are developed for all the GCMs for various rainfall durations. For this purpose, initially, the ratios of intensities of modelled (GCM simulated) to observed (gauged) rainfall for return periods of 2, 5, 10, 25, 50, and 100 years are calculated separately for each GCM for all durations of rainfall. The ratios of the modeled and observed rainfall intensity for various durations at station Kedah for BCC-CSM1.1 under RCP 4.5 are presented in Table 6. Then, the average of these ratios is estimated for each rainfall duration. For theBCC-CSM1.1 model, the average of these ratios estimated at station Kedah for RCP 4.5 is presented in the last line of Table 6. The average of the ratios estimated for other GCMs for RCP 4.5 at this station are given in Table 7.

**Table 6.** Ratios of modelled to observed rainfall for BCC-CSM1.1 for RCP 4.5 at Kedah.

| Return Period (Years) | Duration (Hours) | | | | | | |
|:---:|:---:|:---:|:---:|:---:|:---:|:---:|:---:|
| | 1 h | 3 h | 6 h | 12 h | 24 h | 48 h | 72 h |
| 2 | 0.91 | 0.90 | 1.09 | 1.48 | 1.61 | 1.63 | 1.55 |
| 5 | 1.01 | 1.05 | 1.29 | 1.62 | 1.65 | 1.56 | 1.46 |
| 10 | 1.19 | 1.45 | 1.48 | 1.70 | 1.68 | 1.55 | 1.42 |
| 25 | 1.62 | 1.59 | 1.78 | 1.78 | 1.72 | 1.54 | 1.39 |
| 50 | 2.15 | 1.97 | 2.07 | 1.83 | 1.75 | 1.55 | 1.37 |
| 100 | 2.95 | 2.56 | 2.41 | 1.88 | 1.78 | 1.56 | 1.37 |
| **Average of ratios of Return Period** | 1.64 | 1.59 | 1.69 | 1.72 | 1.70 | 1.57 | 1.43 |

**Table 7.** Average of the ratios of modelled to observed rainfall intensities of return periods 2, 5, 10, 25, 50, and 100 years for various GCMs at Kedah for RCP 4.5.

| Model | Duration (Hours) | | | | | | |
|:---:|:---:|:---:|:---:|:---:|:---:|:---:|:---:|
| | 1 h | 3 h | 6 h | 12 h | 24 h | 48 h | 72 h |
| BCC-CSM1.1 | 1.64 | 1.59 | 1.69 | 1.72 | 1.70 | 1.57 | 1.43 |
| CCSM4 | 1.52 | 1.5 | 1.77 | 1.68 | 1.61 | 1.41 | 1.25 |
| CSIRO-Mk3.6 | 2.17 | 1.98 | 2.09 | 1.96 | 1.85 | 1.61 | 1.44 |
| HadGEM2-ES | 1.41 | 1.49 | 1.75 | 1.91 | 1.9 | 1.76 | 1.6 |
| IPSL-CM5A-MR | 1.67 | 1.66 | 1.91 | 1.81 | 1.71 | 1.52 | 1.35 |
| MIROC-ESM | 2.18 | 1.49 | 1.58 | 1.68 | 1.66 | 1.51 | 1.37 |
| MRI-CGCM3 | 2.81 | 2.68 | 2.86 | 2.8 | 2.75 | 2.51 | 2.27 |
| NorESM1-M | 1.92 | 1.7 | 2.13 | 2.24 | 2.17 | 1.93 | 1.73 |

These ratios are then fitted in their polynomial equations for each GCM to get the values of the model correction factors (MCFs) for all durations of rainfall. Figure 8 shows the MCFs values for various durations of rainfall developed for model BCC-CSM1.1 under RCP 4.5 for station Kedah by putting the average of modelled to observed ratios (taken from Table 7) in its polynomial equation (Equation (7)). The polynomial equations of the GCMs used in current study under RCP 4.5 at station Kedah are given in Table 8. The polynomial equation for BCC-CSM1.1 under RCP 4.5 at Kedah is:

$$y = -0.02\,x^2 + 0.14\,x + 1.47 \qquad (7)$$

where $y$ is the Model Correction Factor (MCF) and $x$ is the average of the ratios of the return periods.

**Table 8.** Polynomials Equations for the GCMs used under RCP 4.5 at Kedah.

| Models | Polynomial Equations | |
|---|---|---|
| BCC-CSM1.1 | $y = -0.02\,x^2 + 0.14\,x + 1.47$ | |
| CCSM4 | $y = -0.04\,x^2 + 0.25\,x + 1.27$ | |
| CSIRO-Mk3.6 | $y = -0.02\,x^2 + 0.04\,x + 2.09$ | |
| HadGEM2-ES | $y = -0.04\,x^2 + 0.38\,x + 1.00$ | where, $x$ is the average of the ratios of return periods |
| IPSL-CM5A-MR | $y = -0.04\,x^2 + 0.23\,x + 1.44$ | and $y$ is the Model correction factor (MCF) |
| MIROC-ESM | $y = 0.02\,x^2 - 0.21\,x + 2.16$ | |
| MRI-CGCM3 | $y = -0.03\,x^2 + 0.18\,x + 2.59$ | |
| NorESM1-M | $y = -0.04\,x^2 + 0.34\,x + 1.47$ | |

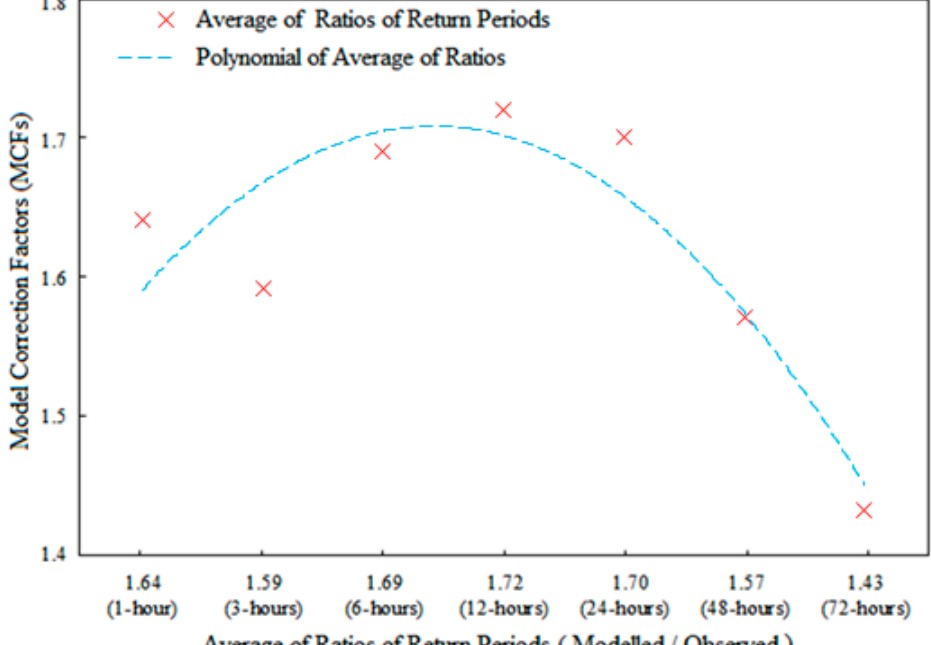

**Figure 8.** Model Correction Factors (MCFs) for modelled rainfall intensities fitted into a polynomial equation for BCC-CSM1.1 for RCP 4.5 at Kedah.

The MCF of a specific duration of rainfall is then multiplied with the return periods of observed rainfall for the same duration of rainfall to generate the return periods for particular rainfall duration in the context of climate change. The MCFs generated for various durations of rainfall at station Kedah under RCP 4.5 for the GCMs used in this study are presented in Table 9. These MCFs are multiplied with the observed return periods for specific duration of observed rainfall intensities separately for each GCM. The rainfall intensities corrected using the MCFs of eight GCMs for various durations, return periods, and RCPs are used for the estimation of the uncertainty of the projected IDF curves of a station.

**Table 9.** Model Correction Factors (MCFs) estimated for different GCMs for RCP 4.5 at Kedah.

| Model | Duration (Hours) | | | | | | |
|---|---|---|---|---|---|---|---|
| | 1 h | 3 h | 6 h | 12 h | 24 h | 48 h | 72 h |
| BCC-CSM1.1 | 1.65 | 1.64 | 1.65 | 1.65 | 1.65 | 1.64 | 1.63 |
| CCSM4 | 1.56 | 1.56 | 1.59 | 1.58 | 1.57 | 1.54 | 1.52 |
| CSIRO-Mk3.6 | 2.08 | 2.09 | 2.09 | 2.09 | 2.10 | 2.10 | 2.11 |
| HadGEM2-ES | 1.46 | 1.48 | 1.54 | 1.58 | 1.58 | 1.54 | 1.51 |
| IPSL-CM5A-MR | 1.71 | 1.71 | 1.73 | 1.73 | 1.72 | 1.70 | 1.68 |
| MIROC-ESM | 1.80 | 1.89 | 1.88 | 1.86 | 1.87 | 1.89 | 1.91 |
| MRI-CGCM3 | 2.86 | 2.86 | 2.86 | 2.86 | 2.86 | 2.85 | 2.84 |
| NorESM1-M | 1.98 | 1.93 | 2.01 | 2.03 | 2.02 | 1.98 | 1.94 |

### 4.6. Development IDF Curves with Uncertainty

The projected rainfall IDF curves under climate change scenarios developed by applying the MCFs of eight GCMs are used to compute the uncertainty level of IDF curves. For this purpose, the intensities of these IDF curves are compared with the intensities of the IDF curves prepared using GCM hindcasts for various rainfall durations to assess the changes in intensities due to climate change. The uncertainty level is the expected upper limit and lower limit of variations in IDF curves due to the changing climate. Using a box and whisker plot, the rainfall intensity values of projected IDF curves for 1st quartile, median, and 3rd quartiles of all eight GCMs are estimated to assess the uncertainty level in rainfall intensity for different durations and return periods. The IDF curves generated with an uncertainty level at station Kedah under RCP 4.5 and RCP 8.5 are shown in Figure 9. First quartile is the lower, while third quartile is the upper uncertainty limit for the projected IDF curves. The maximum and minimum outliers for the GCMs projections are also shown in the figure. These IDF curves show the range of rainfall intensities for various return periods and durations for the period of 2006–2099 with uncertainties under climate scenarios.

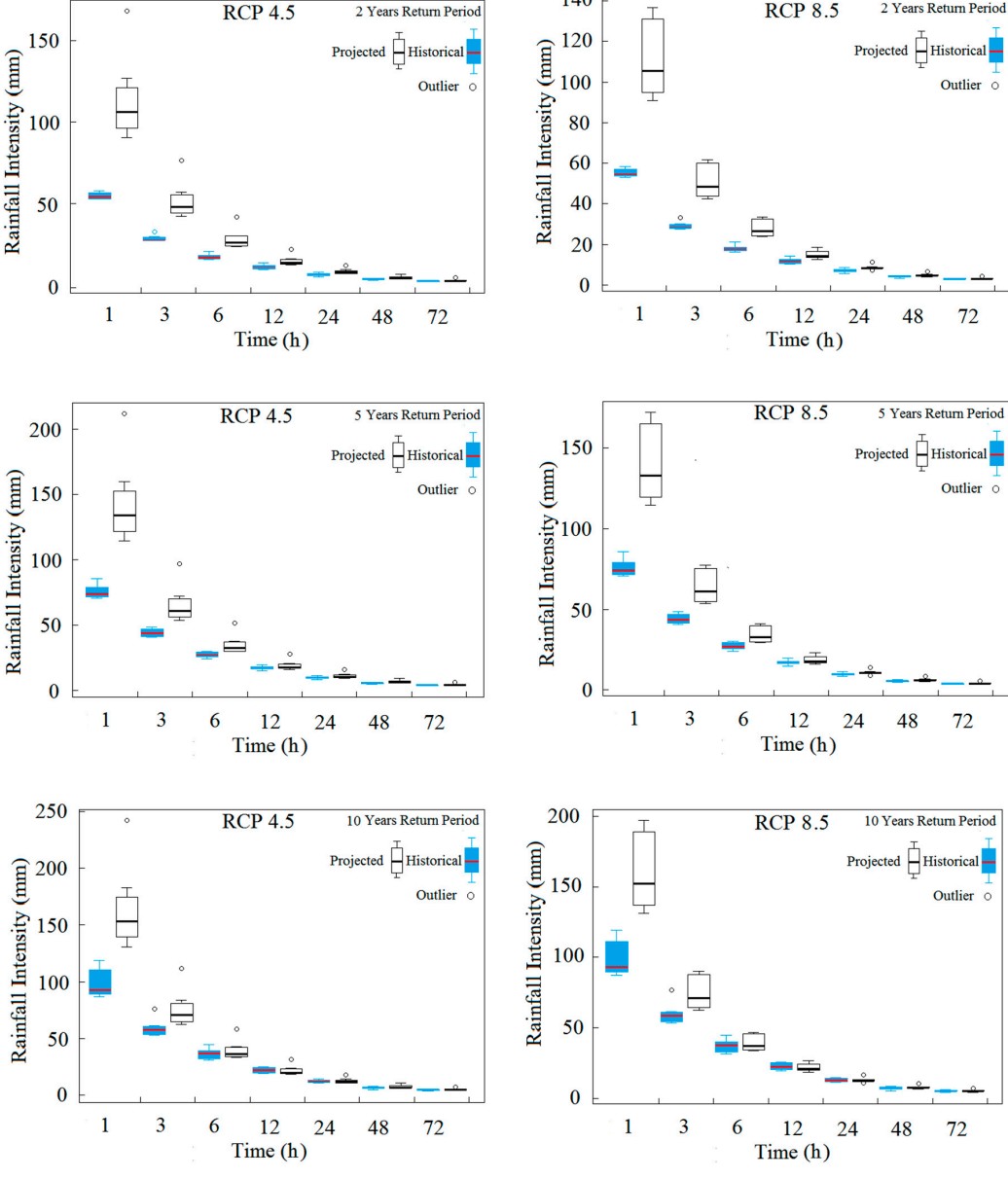

**Figure 9.** *Cont.*

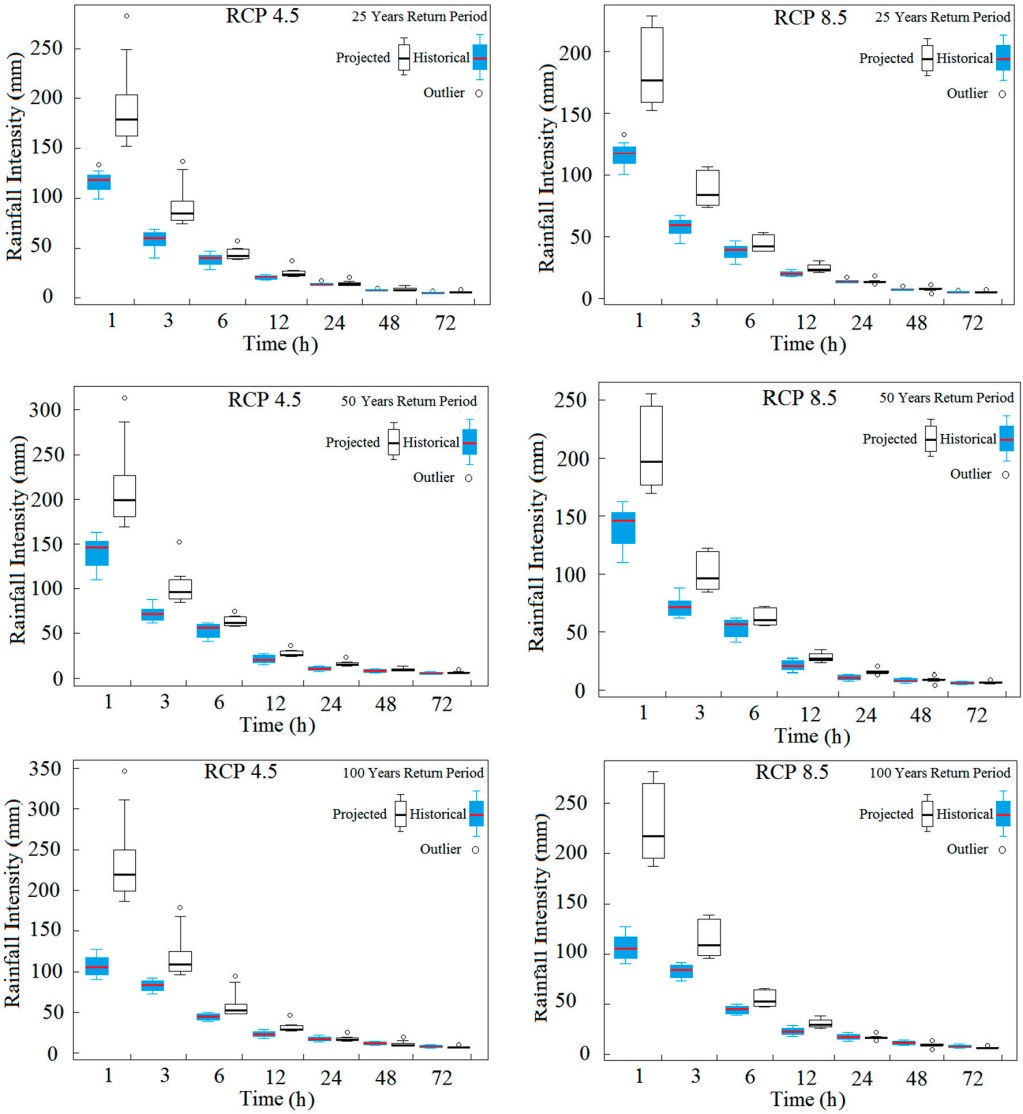

**Figure 9.** Box plot showing the uncertainty in rainfall intensities for different rainfall durations and return periods (2, 5 and 10 years) at station Kedah under RCP 4.5 and RCP 8.5.

## 5. Discussion

The changing climate due to global warming has been found to alter rainfall patterns, making the current urban storm water management infrastructure vulnerable. To minimize losses due to plausible extreme events, the designs for urban infrastructures need to be revised and updated, taking into account the effects of changing climate. It is, therefore, necessary to update the current IDF curves in practice with uncertainty levels in accordance with the climate change projections. This study has been performed to assess the effects of changing climate on rainfall IDF curves over some stations of peninsular Malaysia. The rainfall IDF curves have been developed and updated with the uncertainty under climate scenarios.

Four PDFs and four parameter estimation methods are compared to select the most suitable methods for developing the IDF curves. Based on results of goodness of fit test, GEV is found as the most suitable PDF, while MLE is found to estimate the distribution parameters with least log-likelihood values. It is also observed that log-likelihood estimates do not vary significantly when MLE, GMLE, and L-moments methods are used for estimations of distribution parameters. However, in most cases, MLE is found to estimate the PDF parameters with least log-likelihood values.

Therefore, GEV with the parameter estimation method MLE can be efficiently used for developing IDF curves in Peninsular Malaysia.

GCMs are one of the most important tools for assessing the impact of climate change on rainfall patterns. In this study, non-parametric, distribution-based MOS models are developed for the downscaling of the GCMs projections. The downscaling models often cannot capture extreme rainfall events [53]. However, it is found that the MOS models are capable of accurately capturing rainfall in most of the cases. The higher rainfall is either slightly overestimated or underestimated in some cases. However, the over- or under-estimation is very small, which indicates the capability of QM-based MOS models to downscale daily rainfall.

The variation in rainfall patterns due to the changing climate forces us to update the current IDF curves with expected uncertainty. Therefore, the rainfall IDF curves with uncertainty levels under climate change scenarios are developed by using the projections of eight CMIP5 GCMs under RCP 4.5 and RCP 8.5. Estimation of the effect of climate change on IDF curves reveals greater increases in rainfall intensity for shorter rainfall duration, and the lower increases for longer rainfall duration. The uncertainty level is also found to be higher for lower rainfall durations. For all the GCMs and at all the stations, the uncertainty level is found to be widest for 1 h and lowest for 72 h rainfall duration. This is expected, as the uncertainty in predictions of hourly rainfall is much higher compared to those for daily rainfall. Furthermore, the uncertainty band of higher values is usually higher compared to that of lower values. Therefore, the uncertainty in short-term high intensity rainfall amounts is always much higher compared to that for longer periods of rainfall amounts. This is also evident from the IDF curves prepared using GCM hindcasts (Figure 9). The intensity of shorter duration rainfall is found to increase 2 to 3 times for some GCMs. If the uncertainty level is considered, it is found to increase by 6 times in some cases. This indicates that the higher increase in rainfall intensity of shorter rainfall duration projected by GCMs is highly uncertain.

This study focuses on the development of rainfall IDF curves with uncertainty levels for peninsular Malaysia. The rainfall gauged stations used in the study are mostly covering the urban areas of peninsular Malaysia. However, some stations cover the hilly and forest areas. It is therefore suggested that a greater number of stations be used for better reflection of rainfall patterns in peninsula Malaysia. In this study, eight CIMP5 GCMs under two RCPs are used for projections of future rainfall. More GCMs with all the four RCPs can be used to make efficient assessments of the effects of a changing climate on rainfall IDF curves.

IDF curves are essential for the safe design of hydraulic structures like drainage systems, flood controlling dams, reservoirs, etc. Malaysia often experiences flash floods driven by extreme rainfall which has been projected to increase under climate change scenarios [54]. The IDF curves generated in this study can be used in the design or retrofitting of hydraulic structures in order to adapt to climate change. Increased rates of soil erosion and river sedimentation have been noticed in Malaysia in recent years [55]. Soil erosion is directly linked to rainfall characteristics like intensity and duration [56,57]. The rainfall IDF relationships developed in this study can be used to model soil erosion under climate change scenarios, and for planning soil-erosion prevention practices to mitigate the impacts of climate change. The IDF relationships can be used for the design and calibration of rainfall simulators to make estimations of runoff and soil erosion [58]. The rainfall simulators developed based on the IDF curves generated in this study can be used for the determination of soil erosion susceptibility, and for the planning of soil conservation measures for Malaysia.

## 6. Conclusions

The main objective of the study was to develop IDF curves under climate change scenarios with related uncertainties. A framework was developed to assess the uncertainties in IDF curves under projected climate change scenarios. It can be expected that the IDF curves constructed in this study with best-fit PDF parameters for the return periods of 2, 5, 10, 25, 50, and 100 years can be used for

the planning, design, and operation of hydraulic projects and the efficient management of urban water resources.

The downscaled and observed rainfalls for the historical period are compared to assess the performance of downscaling models. The statistical indices used to measure the performance of downscaling models reveal that non-parametric distribution-based MOS models can efficiently downscale daily rainfall in the peninsular Malaysia. The calibrated MOS models are used for the downscaling of GCM simulated rainfall under RCP 4.5 and RCP 8.5. It is found that the changing climate can cause variation in the rainfall patterns.

The IDF curves are developed using disaggregated daily projected rainfall for the period of 2006–2099. IDF curves are developed for the return periods of 2, 5, 10, 25, 50, and 100-year for the durations of 1, 3, 6, 12, 24, 48, and 72 h. MCFs are used to correct the generated IDF curves for different GCMs. The IDF curves generated for different GCMs are used to develop the IDF curves with uncertainty levels. The results reveal a higher increase in rainfall intensity for shorter durations for the same return periods of rainfall, which gradually decreases for higher durations. The uncertainty in rainfall intensity for different return periods for shorter durations is found to be greater compared to that of higher duration rainfall. It can be concluded that shorter duration rainfall more is uncertain compared to that of higher duration.

Eight GCM simulations under two RCP scenarios are used in this study for the projection of IDF curves with related uncertainties. In future, different ensembles of GCMs can be used to verify the results obtained in the present study. Projections of IDF curves for other RCPs can be generated to show any variation in the uncertainty range.

**Author Contributions:** M.N., and T.I. conceived and designed this study; S.S. analyzed the data, M.N., S.S., J.H.S. and E.-S.C. wrote the paper.

**Funding:** This research was funded by the SeoulTech (Seoul National University of Science and Technology).

**Conflicts of Interest:** The authors declare no conflict of interest.

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
