# Peer review of "Uncertainty in Rainfall Intensity Duration Frequency Curves of Peninsular Malaysia under Changing Climate Scenarios"

_water, doi:10.3390/w10121750_

Reviewer 1 Report

The authors have done a good job in addressing my previous comments and providing few more details where needed. I would just recommend 

(1) One final read from a native speaker to improve the English writing and wording used from time to time.

(2) There is still a cluster of references (7-12) where few individual details (e.g. the country of the research study) should be added

(3) The resolution of the figures should be improved - At least in the version I had access to, they were quite blurry.

Author Response

The authors have done a good job in addressing my previous comments and providing few more details where needed. I would just recommend.

 Comment:

(1) One final read from a native speaker to improve the English writing and wording used from time to time.

Answer: Thank you for your comment. The paper has been proof read by an expert in English Language. All the mistakes have been corrected.

 Comment:

(2) There is still a cluster of references (7-12) where few individual details (e.g. the country of the research study) should be added

Answer: Thank you for your comments. Individual details of the references are given. The revised sentence is as below:

 “Several studies assessed the impacts of climate change in designing urban water management infrastructure in Canada [7-8], Sweden [9], Vietnam [10], United Kingdom [11] and United States  [12].”

 Comment:

(3) The resolution of the figures should be improved - At least in the version I had access to, they were quite blurry.

 Answer: Thank you very much. All the figures are made with a resolution of 300 dpi.

Reviewer 2 Report

I found this to be an interesting and well-written study on an important topic, namely how IDF curves may change under two different global warming scenarios. The authors have crafted a well thought-out procedure for going from the relatively coarse temporal and spatial information in the CMIP5 GCMs to hourly rainfall data at individual stations and made a convincing case that it works. I feel the that manuscript is essentially ready from publication and have only a few minor comments.

I realize it would require a certain amount of additional work, but the conclusions would be greatly strengthened by applying the techniques to at least one additional ensemble member and checking that the conclusions do not change. 

Figure 9 should be recreated using the historical GCM data in place of the observations, including the same estimates of uncertainty. This will not only allow for a more direct assessment of the projected changes (no concerns about model correction factors), but also a sense for the potential spread of the IDF curves in the current climate.

Author Response

Reviewer-2

I found this to be an interesting and well-written study on an important topic, namely how IDF curves may change under two different global warming scenarios. The authors have crafted a well thought-out procedure for going from the relatively coarse temporal and spatial information in the CMIP5 GCMs to hourly rainfall data at individual stations and made a convincing case that it works. I feel the that manuscript is essentially ready from publication and have only a few minor comments.

Comment:

I realize it would require a certain amount of additional work, but the conclusions would be greatly strengthened by applying the techniques to at least one additional ensemble member and checking that the conclusions do not change.

Answer: Thank you very much for your suggestion. Justification of the choice of GCMs and RCP are provided in the manuscript. The results can be checked with an additional ensemble. This has been recommended as future work in conclusion. Following sentences have been added in conclusion section of the revised manuscript for this purpose:

Eight GCM simulations under two RCP scenarios are used in this study for the projection of IDF curves with related uncertainties. In future, different ensembles of GCMs can be used to verify the results obtained in the present study. Besides, projections of IDF curves for other RCPs can be generated to show any variation in uncertainty range.”

 Comment:

Figure 9 should be recreated using the historical GCM data in place of the observations, including the same estimates of uncertainty. This will not only allow for a more direct assessment of the projected changes (no concerns about model correction factors), but also a sense for the potential spread of the IDF curves in the current climate.

 Answer: Thank you for your comment. We revised the figure based on your comment. The related texts are also modified based on revised figure.

Water EISSN 2073-4441 Published by MDPI AG, Basel, Switzerland RSS E-Mail Table of Contents Alert
Back to Top